# A ‘Furry-Tale’ of Zika Virus Infection: What Have We Learned from Animal Models?

**DOI:** 10.3390/v11010029

**Published:** 2019-01-06

**Authors:** Loulieta Nazerai, Jan Pravsgaard Christensen, Allan Randrup Thomsen

**Affiliations:** Department of Immunology and Microbiology, University of Copenhagen, DK 2200 Copenhagen, Denmark; julietanaze@sund.ku.dk (L.N.); jpc@sund.ku.dk (J.P.C.)

**Keywords:** Zika virus, animal models, pathogenesis, immune responses, vaccines

## Abstract

The worldwide attention that the Zika virus (ZIKV) attracted, following its declaration as a Public Health Emergency of International concern by WHO in 2016, has led to a large collective effort by the international scientific community to understand its biology. Despite the mild symptoms caused by ZIKV in most infected people, the virus displays a number of worrying features, such as its ability to cause transplacental infection, fetal abnormalities and vector independent transmission through body fluids. In addition, the virus has been associated with the induction of Guillain-Barre syndrome in a number of infected individuals. With travelling, the virus has spread outside the original ZIKV endemic areas making it imperative to find ways to control it. Thus far, the large number of animal models developed to study ZIKV pathogenesis have proven to be valuable tools in understanding how the virus replicates and manifests itself in the host, its tissue tropism and the type of immune responses it induces. Still, vital questions, such as the molecular mechanisms of ZIKV persistence and the long-term consequences of ZIKV infection in the developing brain, remain unanswered. Here, we reviewed and discussed the major and most recent findings coming from animal studies and their implications for a ZIKV vaccine design.

## 1. Introduction

The Zika virus (ZIKV) is primarily known for its capacity to be transmitted sexually and, as a cause of teratogenesis in the developing fetus [1]. ZIKV, a vector borne-flavivirus, was first isolated in 1947 in the Zika forest of Uganda and for several decades, it was not considered a threatening human pathogen due to the mild and self-limiting symptoms following infection in most people. In 2007, the virus spread outside its original geographical territory and caused a major outbreak on the Micronesian island of Yap where 70% of the population became infected. In 2013, the virus was also recognized to be the cause of a major outbreak in the French Polynesia, and in 2015, ZIKV reached the Americas, where it was rapidly associated with a remarkable increase in numbers of congenital malformations in newborns and Guillain-Barre syndrome in adults. These cases, along with worries that the virus could be sexually transmitted, signaled a significant threat to the whole world and led to the classification of ZIKV as a public health emergency of international concern on February 2016. This classification was withdrawn by the World Health Organization (WHO) on November 2016 and changed to ZIKV being a serious public threat, but not an emergency (Figure 1). Thus far, there have been almost 90 countries and territories around the globe where evidence of vector-transmitted ZIKV infection has been reported.

Prior to the awareness raised in the 21st century regarding the neurologic complications accompanying ZIKV infection, the ZIKV literature did not exceed 40 published reports. In these early reports, it was obvious that adult mice were somehow ‘resistant’ to peripheral ZIKV infection. Today, the increased effort to understand the special features of ZIKV has led to the development of a series of animal models (Figure 2), which support ZIKV replication and has resulted in around 2900 published papers (as of October 2018), a number that keeps increasing day by day.

Mouse models: Being the most commonly used laboratory animals, mice were the first to be exploited for modeling ZIKV infection. The first challenge was to enable viral replication in adult immunocompetent mice, which, under normal circumstances and unless the administered inoculum is extremely high, do not exhibit clinical symptoms of the viral disease. Unlike the situation in humans, where ZIKV can successfully replicate by evading the type I interferon (IFN) response, due to species-specific evasion mechanisms, ZIKV replication in mice is not sustainable. Therefore, adult immunocompromised mice carrying genetic deficiencies in the IFN-pathway (*Ifnar1* knockout, *Ifngr1* knockout, *Stat2* knockout, *Irf3/Irf5* double knockout, *Irf3/Irf5/Irf7* triple knockout) and neonatal mice known for their limited type I IFN responses, have been extensively applied [2,3,4,5,6,7,8]. These models have been valuable tools in delineating ZIKV infectious cycle, cell tropism and transmission as well as in the evaluation of vaccine and drug candidates. However, the primary limitation of such models is that they do not allow for a comprehensive study of ZIKV immunity due to an absent or weak first line of defense, which is critical for the antiviral response. To amend that, other mouse models have been established, where the innate immune responses are present but are either temporarily suppressed by the administration of anti-IFNAR abs prior to ZIKV exposure [9,10,11] or circumvented via an alternative route of virus administration [12]. Additionally, transgenic mice where the mouse *Stat2* was exchanged with human *STAT2* to allow ZIKV susceptibility [13] and a BLT humanized mouse model with a transplanted human immune system to allow the assessment of human specific immune responses have been suggested [14].

Non-human primate (NHP) models: A wide range of large animals, including goats, sheep, water buffalos, lions and NHPs, have been found to elicit an immune response upon ZIKV infection [15]. Nevertheless, the resemblance of NHPs’ physiology to that of humans has rendered them the main large animal model for studying ZIKV infection. Rhesus (*Maccaca mulatta*) macaques, and to a lesser extent pigtail (*Maccaca nemestrina*) and cynomolgus (*Maccaca fascicularis*) macaques, have been employed to study the consequences of primary ZIKV infection in terms of clinical symptoms and viral tropism. Infection of macaques with different ZIKV strains has indicated that, despite the absence of clinical symptoms in some cases, viremia may last for several weeks depending on ZIKV strain and macaque species [16,17]. In addition, macaques have been extensively used for understanding the protective immunity and evaluation of ZIKV vaccine candidates [17,18]. Nevertheless, olive baboons (*Papio Anubis*) are probably closer to humans than any other NHP in terms of size, genetics and immune repertoire and therefore, have been proposed as a more translationally relevant model for ZIKV research [19]. New World primates, including the marmoset (*Callithrix jacchus*), squirrel monkey (*Samiris sciureus*) and owl monkey (*Aotius lemurinus)*, have also been shown to support ZIKV replication and exhibit persisting viremia in the absence of clinical symptoms [20,21]. There is no doubt that NHP models can allow for a more accurate assessment of ZIKV pathogenesis, and provide valuable insight into safety and efficacy of vaccines and drugs as well as their optimal dose and administration route. Fetal development and placental structure in NHPs bear greater similarity to humans than in the case of other animal models; however, the duration of the gestational period is substantially shorter (20–26 weeks in NHPs versus 40 weeks in humans). Still, data produced in pregnant NHP models come at a much slower rate compared to small animal models. On top of that, NHPs are costly to purchase, house and maintain, which can limit the number of animals used, and therefore, affect the statistical power of each study. More importantly, the ethical issues represent a greater controversy for NHPs than for small animal models, even in the case of non-endangered species [22,23].

Other models: Less popular models like guinea pigs, hamsters and chicken embryos have also been developed for ZIKV research. Guinea pigs are susceptible to ZIKV both when infected subcutaneously and intranasally. The latter allows modeling of close contact transmission of ZIKV in humans [24]. Additionally, the reproductive physiology of guinea pigs is similar to humans and, in contrast to other rodent pups, guinea pig pups are born with a mature central nervous system (CNS)—a property that could be explored to study the neurological manifestation of ZIKV in infants [25]. Both adult and fetal hamsters can support ZIKV replication and are severely infected, especially when *Stat2* is knocked out [26]. Chicken embryos can be successfully infected by ZIKV on days E2.5 or E5, and while the infection is not lethal, it can cause a microcephaly-like phenotype at later stages of development (E15 and E20). These findings suggest that chicken embryos can offer an alternative to study mechanisms of ZIKV infection in the developing nervous system of the fetus [27].

## 2. Immune Response and Immune Evasion

Mosquitos of the Aedes genus are the primary carriers of ZIKV [28]. When an infected mosquito bites a healthy individual, it will go with its proboscis through the epidermis and reach for a venule. In that process, saliva containing the infectious virus is released locally but also into the bloodstream. Local dendritic cells are the first to be infected, and will subsequently migrate to the draining lymph nodes to initiate an adaptive immune response [29]. Presumably, ZIKV enters its target cells via receptor mediated endocytosis and releases its 11 kb, single-stranded, positive-sense RNA genome [30]. Translation of the viral RNA generates one polyprotein that is subsequently catalytically processed into 10 mature proteins: three structural (C, prM/M and E) and seven non-structural (NS1, NS2A, NS2B, NS3, NS4A, NS4Band NS5) [31]. The infectious capacity of ZIKV is highly dependent on the prM, E, NS1, NS3 and NS5 proteins. Interestingly, while the sequences of ZIKV NS3 and NS5 proteins are highly similar to other flaviviruses, the sequences of prM, E and NS3 are significantly different [32]. This might explain why, while ZIKV shares around 80% similarity at the molecular level with the rest of the flaviviruses, it displays such distinct features in terms of cellular tropism, transmission and persistence. Nevertheless, the mechanisms that ZIKV employs to avoid immune recognition by the host are similar to those of other flaviviruses (i.e Dengue, West Nile and Japanese Encephalitis virus) and include blockade of IFNs, natural killer (NK) cells, complement system, B and T cell responses [33,34]. Immune evasion is mediated by the non-structural proteins, especially NS1 and NS5, which act collectively to limit and escape host antiviral responses via interference with critical signaling pathways [35].

The initial recognition of RNA viruses (mediated via recognition of pathogen-associated molecular patterns by host pattern recognition receptors), turns on the type I IFN signaling pathway and results in the secretion of type I and type III IFNs. The engagement of the produced IFNs to their receptors (IFNAR1/IFNAR2 and IFNLR1/IL10Rβ, respectively) activates the Janus kinase/signal transducers and activators of transcription (JAK/STAT) pathway and causes the subsequent upregulation of interferon-stimulated genes (ISGs). In humans, the ZIKV NS5 protein binds to STAT2 and degrades it, and in this way the virus manages to disrupt the type I IFN signaling pathway. However, in mice, ZIKV NS5 is unable to bind efficiently to STAT2 and the virus is hence unable to escape innate immune recognition [36,37,38]. Therefore, to recapitulate human disease in mice, mouse models with genetic or acquired deficiencies in the IFN pathway have been extensively employed [39].

Innate immune responses are vital in controlling ZIKV infection, which is highlighted by the necessity to evade them to make mice susceptible [40]. Type I IFN deficient/suppressed mice are permissive to ZIKV replication from the age of 3 weeks and up to 6 months, while the selection of the right mouse strain as well as the proper virus strain and administration route are critical for achieving uniform lethality [7,39,41]. Based on studies of such immunocompromised mice, but also other rodent models, it is evident that following infection, ZIKV can reach multiple organs and may persist long term in immune-privileged sites (i.e., brain, eyes, female/male reproductive tract). In addition, the virus can be found in several body fluids (i.e., saliva, tears, semen and urine) [26,42]. Interestingly, there is also evidence pointing to a pathogenic role of innate recognition. Thus, in mice, type I IFN has been found to play a significant role in the fetal demise following intrauterine infection [8].

Adaptive immune responses are also essential in restricting ZIKV replication [43]. During ZIKV infection, the induced antibodies are primarily directed against the structural proteins envelope (E) and pre-membrane (prM) as well as the intracellularly secreted non-structural 1 (NS1) protein, which makes them good vaccine targets [44,45]. The T cell response is primarily directed against E, prM, NS3 and NS5 proteins with the immunodominant peptides being highly conserved amongst different ZIKV strains [46]. Yet, while primary asymptomatic ZIKV infection of WT immunocompetent adult mice induces both the humoral and cellular arm of adaptive immunity, both arms are not equally important for resistance to re-infection. The induction of anti-ZIKV antibodies is the most essential for viral control since adoptive transfer of immune serum, but not immune splenocytes, was able to save mice from lethal intracerebral (i.c.) infection with ZIKV [12]. In support of the central role of neutralizing antibodies (nAbs), protective immune responses against the ZIKV protein E or even the EDIII fragment alone were found to be sufficient to prevent lethality in pups and type I IFN deficient mice [47,48]. Overall, the results regarding the contribution of humoral responses in protection are conclusive and suggest a key role for the induced nAbs. The contribution of T cells on the other hand, and in particular the role of CD8 T cells in protection and immunopathology, has been more debated. Studies in IFNAR-suppressed Rag KO and LysMcre + IFNARfl/fl mice suggest a substantial role of CD8 T cells, while studies in WT immunocompetent adult mice suggest that their presence primarily serves as a back-up mechanism when the levels of nAbs are not sufficient [12,43,46]. This is supported by recent preliminary data suggesting that in type I IFN-replete mice, CD8 T cells may provide significant early antiviral activity, provided they are present in high numbers (personal unpublished observation). In one study of type I IFN deficient mice, evidence indicated that virus-specific CD8 T cells not only contribute to viral control, but also caused hindlimb paralysis [49]. Whether this is also the case in type I IFN replete mice is not clear. However, in personal unpublished experiments where we compared the clinical course of i.c. infection in RAG-deficient mice, CD8 deficient mice and wildtype controls, we did not observe any obvious difference in symptomatology, suggesting that perhaps immune mediated induction of paralysis was special to mice lacking type I IFN signaling, possibly as a consequence of increased viral dissemination under these conditions. On the other hand, CD4 T cells, given their role in proper humoral responses, have been found to be critical for protection both in immunocompetent and immunocompromised mouse models [12,50,51].

## 3. Transmission

The primary route of ZIKV transmission is via the bite from an infected mosquito. In addition, unlike most flaviviruses, ZIKV can spread both vertically, through transplacental transmission, and horizontally, through sexual transmission, but potentially also through close contact with viremic individuals.

### 3.1. Transplacental Transmission

Pregnant women comprise the most vulnerable group in terms of ZIKV infection due to the virus’s documented ability to cross the placental barrier and reach the fetus. Microcephaly is the most severe and noticeable consequence of fetal congenital infection; nevertheless, intrauterine ZIKV exposure can also affect the developing brain in less obvious ways causing neurological irregularities that can impair proper mental and physical development later on in life. Studies on both immunocompetent and immunocompromised pregnant mouse models, but also pregnant NHPs, confirm that ZIKV can compromise placental integrity and that the time of infection during pregnancy correlates with the severity of congenital infection in the fetus [8,11,52,53,54]. Interestingly, infection of a pregnant pigtail macaque with ZIKV caused no apparent clinical disease in the mother, but was able to cause serious damage in the fetus, suggesting that even in asymptomatic pregnant women the virus may still reach the fetal brain [53]. ZIKV seems to be able to infect almost all brain cell types and can induce apoptosis in neuronal progenitor cells (NPCs) during development, in this way interfering with normal brain growth in the embryo [55]. The basis of ZIKV neuronal tropism is still not clear; nevertheless, the virus seems to reach the CNS without damaging the blood-brain barrier [56]. One of the strategies of the virus to establish its replication is via hijacking RNA-binding proteins that are abundant in NPCs. Interestingly, the RNA-binding protein musashi 1 (MSI1) is absent from mature neurons, but is vastly expressed in NPCs as well as in several other ZIKV susceptible tissues [57]. Once in the CNS, ZIKV is able to interfere with cellular pathways critical for fetal brain development, and, as shown in an immunocompetent mouse model hosting human *STAT2*, ZIKV NS4B protein limits IFN-β production and hence inhibits the AKT-mTOR signaling cycle [13].

### 3.2. Sexual Transmission

Sexual transmission is a unique and particularly worrying feature of ZIKV infection because it can contribute to viral spread outside the ZIKV endemic areas. The first confirmed case of sexual transmission dates back to 2008, when an American scientist got infected with ZIKV while working in Senegal, and transmitted the virus to his wife when he returned home [58]. Since then, more cases of ZIKV transmission in non-endemic regions have been reported, with the majority of them being transmissions from male to female partners. The presence of viral RNA in the semen of infected men up to 6 months following primary infection supports sexual transmission, while it also indicates the window of infectivity [59]. On top of that, the long-term persistence of ZIKV in the testes and semen of male mice and NHPs explains why sexual transmission is favored [60,61,62]. The specific cell types of the reproductive tract that function as a reservoir for ZIKV are not clear yet; nevertheless, the virus seems to be able to infect almost all components of the male and female reproductive tract [8,63,64]. Male to female transmission has been observed in mouse models, the opposite, however, was unsuccessful [62]. Interestingly, female mice experienced a more severe disease when infected sexually compared to infection via other routes, indicating that sexual transmission can potentially alter viral tropism and enhance viral dissemination [65]. Infertility, as a result of ZIKV infection, was observed in male mice where viral presence in the testes caused testicular atrophy [66]. However, it is less clear how ZIKV affects female infertility.

### 3.3. Close Contact Transmission

The risk of ZIKV transmission through direct contact with infectious body fluids/contaminants has not been studied extensively. Notably, the potential of close contact transmission has been illustrated for other members of the flavivirus family like the JEV, which can be transmitted via oronasal secretions in pigs, and WNV, which can be transmitted via direct contact amongst geese [67]. Nasal infection of mice and NHPs with ZIKV can establish a successful systemic infection. The capacity of ZIKV to be transmitted via close contact is further underscored by the fact that naive guinea pigs can get infected just by housing them with ZIKV-infected guinea pigs [68]. In agreement with those observations, a case of ZIKV infection via a non-mosquito and non-sexual route in a human has been reported [69]. Further research is needed to elucidate the mechanisms and the conditions under which close contact with ZIKV contaminants facilitates transmission.

## 4. Evolution

After its initial isolation in 1947 in Africa, ZIKV remained relatively friendly for several decades before it reached the Americas. Phylogenetic analysis has shown that ZIKV has evolved into two distinct lineages: the East/West African (MR766/Nigerian cluster) and the Asian lineage, which had been silently circulating in Africa and Asia [70]. Strains belonging to the Asian lineage are linked to the outbreaks in Micronesia (2007), French Polynesia (2013) and the southeast of America (2015). The increase in the infectious capacity of the virus and its new adverse properties suggest that ZIKV underwent some molecular and genetic changes throughout the years. Indeed, amino-acid analysis of pre- and post-epidemic ZIKV variants has revealed the accumulation of amino-acid changes on the viral glycoprotein and in particular the E, prM, NS1, NS2A and NS5 ZIKV proteins [71,72]. Remarkably, an amino acid substitution at residue 139 in the prM protein, from serine (S) to asparagine (N), coincides very accurately with the initiation of the epidemic in 2013 [73]. That substitution has been maintained during the subsequent journey of the virus in the Americas. In a pregnancy mouse model, it was confirmed that the single point mutation at residue 139 in the prM region can enhance the severity of microcephaly in fetal mouse [74]. On top of that, a single point mutation at residue 188 in the NS1 ZIKV protein has been shown to substantially increase infectivity when injected in immunocompromised mice and enhance neurovirulence when injected directly in neonatal mouse brain [72,75]. In spite of the genetic variations, there is currently only one ZIKV serotype [76]. Nevertheless, the high mutation rates that characterizes many RNA viruses raises the concern that new serotypes may emerge, which can impend the development of a universal protective vaccine.

## 5. Vaccines

Vaccines are essential for controlling viral spread. There are currently no licensed vaccines to prevent ZIKV infection; however, several candidates have reached clinical trials. These include DNA, mRNA and vector-based vaccines encoding the prM/E proteins as well as purified inactivated ZIKV vaccines (Figure 3) [18,77,78,79,80,81,82,83,84,85,86,87]. Before reaching clinical trials, those vaccines were evaluated in animal models and displayed a protective capacity that was clearly associated with high levels of nAbs. The contribution of cellular immune responses seems not to be critical for protection when there are sufficient levels of nAbs against the E protein [12,79]. In fact, the adenovirus (Ad), recombinant DNA (rDNA) and virus-like proteins (VLPs) displaying only domain III of the E protein were successful in inducing potent nAbs in mice [47,88,89]. The other two domains (EDI, EDII) of the E protein showed a lower neutralizing potential, which raises concerns for their contribution to antibody-dependent-enhancement (ADE) [45]. In order to avert the risk of ADE, alternative vaccine targets have been exploited with the NS1 ZIKV protein being quite promising. When NS1 is delivered with the Modified Vaccinia Ankara (MVA) vector, it can confer full protection in immunocompetent mice against lethal i.c. challenge [90]. Moreover, when delivered with the recombinant Vesicular Stomatitis Virus (rVSV) vector, NS1 can confer partial protection in immunocompromised mice [91]. Interestingly, combining NS1 with prME has enhanced the in vivo performance of rVSV- and Ad2-based vaccines in mouse models [91,92]. Efforts on vaccine development are still intense and focus on finding alternative delivery platforms and on exploring the optimal antigen combinations to induce safe and efficacious protection.

## 6. Conclusions

Modeling human viral diseases in animal species is an indispensable part of studying and understanding disease pathophysiology. For ZIKV, the large number of animal models developed has contributed greatly to our understanding of this virus and protective immunity. Still, the challenge lies in selecting the right model to address the right scientific question properly. Mouse models are the frontrunners in delineating the molecular mechanisms underlying ZIKV infectivity and the preliminary evaluation of vaccines and therapeutics in a cost-effective manner. Nevertheless, there is currently no mouse model to provide uniform lethality and the levels of susceptibility often rely on the mouse and virus strain selected as well as the time and route of viral administration.

## Figures and Tables

**Figure 1 viruses-11-00029-f001:**
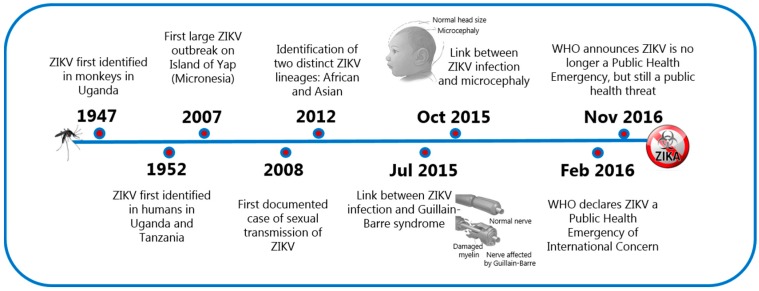
The chronology of the most crucial events of the Zika virus (ZIKV) infection. Images adapted under license from Shutterstock.com.

**Figure 2 viruses-11-00029-f002:**
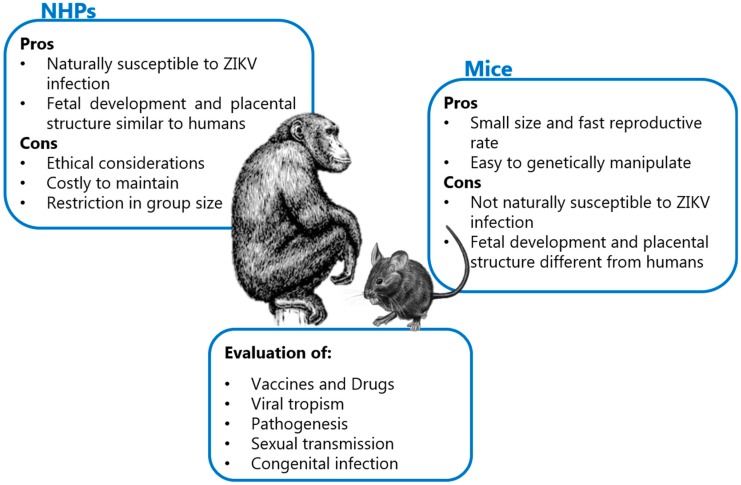
Overview of the advantages, disadvantages and the contribution mice and non-human primates (NHPs), which are the most widely used animal models for ZIKV infection. Other models, such as guinea pigs, hamsters and chicken embryos, are also available but less popular. Images adapted under license from Shutterstock.com.

**Figure 3 viruses-11-00029-f003:**
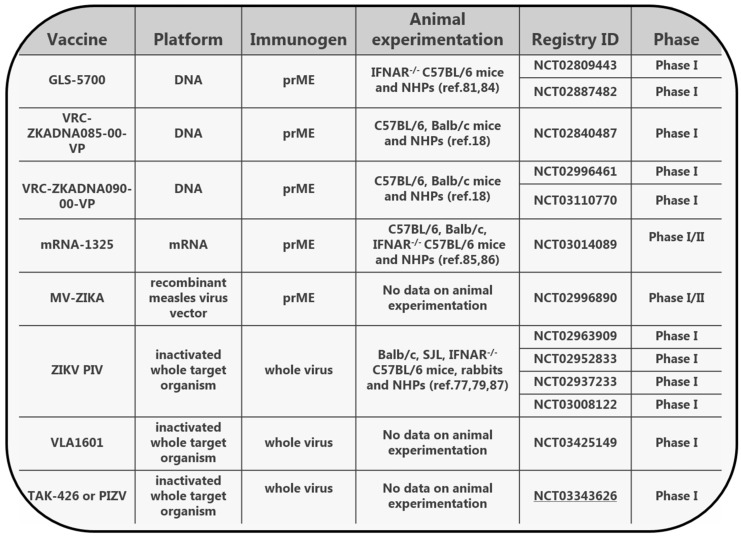
Overview of the ZIKV vaccines currently in clinical trials. Source: World Health Organization (WHO) vaccine pipeline tracker (https://www.who.int/immunization/research/vaccine_pipeline_tracker_spreadsheet/en/).

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
