# Peer review of "A ‘Furry-Tale’ of Zika Virus Infection: What Have We Learned from Animal Models?"

_viruses, 2019, doi:10.3390/v11010029_

Reviewer 1 Report

This review by Nazerai et al was a very comprehensive article on the use of animal models for ZIKV biology. The authors covered the types of models out there, pros and cons, and how they have been used. It was clear, concise, and easy to read.

My only one comment for content is in the "Immune response and Immune evasion" section. There are several studies on the role the immune response in causing ZIKV induced pathology that came from animal studies. These include the papers Jurado KA et al, 2018 and Yockey LJ et al, 2018. I think it would be nice to address the adverse effects of the immune response as well as the beneficial ones for clearance. There was also a recent paper published on CD4 cell and the antibody response (Lucas CGO et al, 2018). This paper should be included in this section.

Author Response

My only one comment for content is in the "Immune response and Immune evasion" section. There are several studies on the role the immune response in causing ZIKV induced pathology that came from animal studies. These include the papers Jurado KA et al, 2018 and Yockey LJ et al, 2018. I think it would be nice to address the adverse effects of the immune response as well as the beneficial ones for clearance. There was also a recent paper published on CD4 cell and the antibody response (Lucas CGO et al, 2018). This paper should be included in this section.

-These papers have now been cited and the findings of the studies have been discussed.

Reviewer 2 Report

This is a brief review of how animal models have contributed to current knowledge of ZIKV tropism, pathogenesis and vaccine strategies. In general, the review is accurate and clearly written. However, with regard to NHPs, the information is a bit too generalized. Additionally, a few minor revisions should be included.

1.    The section on NHP models is relatively brief, and tends to lump all NHPs into a single group. Although rhesus macaques have been used in the majority of studies, other macaques (pigtail, cynomolgus) have been used, as well as other NHP species (marmosets, baboons). Similar as well as contrasting findings of these models should be noted. 

2.    Since this review is focused on animal models, the vaccine data in table 1 should include a column noting the nature and results of animal experiments with these vaccines, or this could be discussed in more detail in the text. 

3.    Line 79: There are, of course, significant ethical considerations for NHP experimentation. However, the species used for ZIKV experiments are primarily purpose-bred for research, so that experimentation does not impact/ threaten wild populations.

4.    Line 157: The statement about ZIKV being unique in its ability to spread via transplacental transmission is not entirely accurate. Although maternal-fetal transmission of ZIKV received much more attention, largely due to the large outbreak and dramatic phenotype in some infected neonates, other flaviviruses have been demonstrated to transmit vertically during pregnancy. (See Charlier, et al (2017) Lancet Child and Adolescent Health 1:134; PMID:30169203 for review). 

5.    In a few instances, an abbreviation (e.g. nAbs) is given, with the full phrase in parentheses (neutralizing antibodies). Usually, the phrase is spelled out with the abbreviation in parentheses, and the abbreviation used after that (as with “Zika virus (ZIKV)” in line 25). 

Author Response

1.    The section on NHP models is relatively brief, and tends to lump all NHPs into a single group. Although rhesus macaques have been used in the majority of studies, other macaques (pigtail, cynomolgus) have been used, as well as other NHP species (marmosets, baboons). Similar as well as contrasting findings of these models should be noted. 

-The section has been extended. Nevertheless the purpose of the paragraph is to give the general overview of the NHPs used. Specific findings and experiments along with the relevant citations are mentioned throughout the text.

2.    Since this review is focused on animal models, the vaccine data in table 1 should include a column noting the nature and results of animal experiments with these vaccines, or this could be discussed in more detail in the text. 

-There is now an additional column on table 1 about ‘animal experimentation’ with relevant references.

3.    Line 79: There are, of course, significant ethical considerations for NHP experimentation. However, the species used for ZIKV experiments are primarily purpose-bred for research, so that experimentation does not impact/ threaten wild populations.

-Indeed, NHP experimentation does not always threaten wild populations. But we still find that there are ethical considerations regarding their use, not least in the view of the public (e.g. our own university does not allow NHP experiments even though the national regulations allows them).

4.    Line 157: The statement about ZIKV being unique in its ability to spread via transplacental transmission is not entirely accurate. Although maternal-fetal transmission of ZIKV received much more attention, largely due to the large outbreak and dramatic phenotype in some infected neonates, other flaviviruses have been demonstrated to transmit vertically during pregnancy. (See Charlier, et al (2017) Lancet Child and Adolescent Health 1:134; PMID:30169203 for review). 

- We agreed with the reviewer and the related sentences have been modified.

5.    In a few instances, an abbreviation (e.g. nAbs) is given, with the full phrase in parentheses (neutralizing antibodies). Usually, the phrase is spelled out with the abbreviation in parentheses, and the abbreviation used after that (as with “Zika virus (ZIKV)” in line 25). 

-The abbreviations have been corrected.